# Identification of Novel Coloboma Candidate Genes through Conserved Gene Expression Analyses across Four Vertebrate Species

**DOI:** 10.3390/biom13020293

**Published:** 2023-02-04

**Authors:** Violeta Trejo-Reveles, Nicholas Owen, Brian Ho Ching Chan, Maria Toms, Jeffrey J. Schoenebeck, Mariya Moosajee, Joe Rainger

**Affiliations:** 1Roslin Institute, R(D)SVS, Easter Bush Campus, University of Edinburgh, Edinburgh EH25 9RG, UK; 2Development, Ageing and Disease, UCL Institute of Ophthalmology, London EC1V 9EL, UK; 3Ocular Genomics and Therapeutics, The Francis Crick Institute, London NW1 1A, UK; 4Department of Genetics, Moorfields Eye Hospital NHS Foundation Trust, London EC1V 2PD, UK

**Keywords:** coloboma, microphthalmia, anophthalmia, optic fissure closure, eye development

## Abstract

Ocular coloboma (OC) is a failure of complete optic fissure closure during embryonic development and presents as a tissue defect along the proximal–distal axis of the ventral eye. It is classed as part of the clinical spectrum of structural eye malformations with microphthalmia and anophthalmia, collectively abbreviated to MAC. Despite deliberate attempts to identify causative variants in MAC, many patients remain without a genetic diagnosis. To reveal potential candidate genes, we utilised transcriptomes experimentally generated from embryonic eye tissues derived from humans, mice, zebrafish, and chicken at stages coincident with optic fissure closure. Our in-silico analyses found 10 genes with optic fissure-specific enriched expression: *ALDH1A3*, *BMPR1B*, *EMX2*, *EPHB3*, *NID1*, *NTN1*, *PAX2*, *SMOC1*, *TENM3*, and *VAX1*. In situ hybridization revealed that all 10 genes were broadly expressed ventrally in the developing eye but that only *PAX2* and *NTN1* were expressed in cells at the edges of the optic fissure margin. Of these conserved optic fissure genes, *EMX2*, *NID1*, and *EPHB3* have not previously been associated with human MAC cases. Targeted genetic manipulation in zebrafish embryos using CRISPR/Cas9 caused the developmental MAC phenotype for *emx2* and *ephb3*. We analysed available whole genome sequencing datasets from MAC patients and identified a range of variants with plausible causality. In combination, our data suggest that expression of genes involved in ventral eye development is conserved across a range of vertebrate species and that *EMX2*, *NID1*, and *EPHB3* are candidate loci that warrant further functional analysis in the context of MAC and should be considered for sequencing in cohorts of patients with structural eye malformations.

## 1. Introduction

Congenital structural eye malformations can cause severe visual impairment and reduced quality of life, with a prevalence ranging up to 19/10,000 newborns [1,2,3]. However, the genetic and environmental causes remain elusive, leading to a lack of clinical management or treatment options. Within these disorders, ocular coloboma (OC) is the most common and considered part of the clinical phenotypic spectrum, along with anophthalmia (a completely missing eye) and microphthalmia (a small, underdeveloped eye), together referred to as MAC. The developmental insults that cause MAC are mostly genetic but may also be influenced by environmental factors [4,5,6]. They occur early in gestation and typically affect the morphogenesis and structural development of the optic vesicle (anophthalmia) or eye cup (microphthalmia and coloboma) [7]. During normal eye development, the anterior neural plate develops a transcriptionally distinct eye field that evaginates bilaterally from the surrounding neuroepithelium as the neural plate folds [8]. These outgrowing neural pouches are referred to as the optic vesicles, and each one eventually reaches the overlying surface ectoderm, triggering a mutual invagination of the two tissues to create a bilayered optic cup and lens vesicle [9,10].

The ventral retina remains undifferentiated during the optic cup stages, largely through a unique transcriptional signature that enables this region of the eye to undergo morphogenic changes that distinguish it from the dorsal retina [11,12,13,14,15]. The morphogenesis of the ventral eye creates a gap along the proximal-distal axes of each optic cup—the optic fissure. The edges of the optic fissure come closer together as the optic cups continue to grow, displacing neural-crest-derived periocular mesenchyme (POM) and vascular endothelia, until they eventually come into direct contact and fuse together to complete the circumferential continuity of the eye. The genetic and physical processes that control the fusion of these opposing epithelial sheets are poorly understood but require the decoupling and reorientation of cells at the leading edges of the epithelia, similar to a partial epithelial to mesenchymal transition, and coincident with remodelling of overlying basement membrane structures [16,17,18].

Clinically, OC manifests as an inferonasal gap that could affect one or more tissues, including the iris, ciliary body, neural retina, retinal pigmented epithelium (RPE), and choroid, with/without involvement of the optic disc and macula. It accounts for 10% of childhood blindness and has a variable prevalence, but may affect up to 11.9 per 100,000 births [2,19]. It can manifest with wider systemic features as part of a syndrome (syndromic OC), be associated with other eye defects (complex OC), or occur in isolation (isolated OC, and can affect only one (unilateral) or both (bilateral) eyes.

The use of next-generation sequencing, including targeted gene panels and whole genome sequencing, in MAC patients and their families has led to an increase in genetic diagnosis rates of up to 33% [20], with MAC panels now including up to 86 different genes [21]. However, very few OC loci show recurrence among unrelated families, and recent studies suggest that over 80% of cases do not have a genetic diagnosis [7,22,23].

Animal models of optic fissure closure (OFC) have recently been powerful in revealing the transcriptional landscapes associated with ventral eye development and fissure fusion, successfully identifying novel MAC candidate genes [17,24,25]. Using these datasets, we previously generated a pipeline to enable the identification of new coloboma candidate genes based on evolutionary conserved gene expression in the developing optic fissure [26]. This method leveraged microarray data from mice [24] and RNAseq data from zebrafish [25] to reveal four novel coloboma candidate genes and supported the identification of human pathogenic variants therein [26]. Here, we applied a similar strategy, this time exclusively using RNAseq data but also increasing the analysis to four vertebrate species, representing mammals (humans and mice), fish (zebrafish), and avians (chickens). Our analyses revealed three novel candidate genes for roles in eye development, with supporting evidence from in vivo work that these are essential for OFC. Based on this, we propose that *NID1*, *EMX2*, and *EPHB3* are now included in targeted gene panels for MAC patients and their families.

## 2. Materials and Methods

### 2.1. Chicken Eye mRNA Sequencing

RNAseq was performed on dissected tissue collected from dorsal and fissure regions of the chicken eye at both the fusing (HH St30) and fully fused (HH St34) stages of optic fissure closure (as described in Hardy et al., 2019 [17] and staged according to Hamburger and Hamilton criteria [27]). Three replicates for each sample stage and region were prepared, and each sample replicate contained pooled tissue from a minimum of 10 different embryos. Pooled samples were processed for RNA extraction on the day of dissection. Total RNA was extracted using Trizol reagent (Ambion) and treated with Turbo DNaseI (Thermo #AM2238). Libraries were prepared from 500 ng of the total RNA sample using the NEBNEXT Ultra II Directional RNA Library Prep kit (NEB #7760) with Poly-A mRNA magnetic isolation (NEB #E7490) according to the manufacturer’s protocol. mRNA libraries were assessed on the Agilent 2100 Electrophoresis Bioanalyser with the DNA High Sensitivity Kit (#5067-4627) for fragment size distribution and then quantified using the Qubit 2.0 Fluorometer and the Qubit dsDNA HS assay kit. Libraries were run on the Nextseq 550 using a high output flow cell and generated a cluster density of 227 K/mm^2^, with 86% of clusters passing quality filters (PF). This produced 83.1 Gb of data with 89.2% ≥Q30. Coverage for each of the mRNA libraries generated ≥34 M PE (2× 75 bp) reads (min: 34 M, max: 43 M, mean: 40 M).

### 2.2. Transcript Quantification and Differential Gene Expression Analysis

Sequencing quality was evaluated using FASTQC (version 0.11.4). Sequencing adaptors were trimmed using Cutadapt [28] (version 1.16) and Trimmgalore (version 0.4.1). Trimmed sequences were aligned to the chicken genome (galgal6) using STAR aligner to generate bam files, which were then coordinate sorted and indexed using Samtools [29] (version 1.6). A count matrix of transcripts annotated in Ensembl (version 103) was subsequently generated from bam files using featureCounts, part of the Subread package [30,31] (version 2.0.1). Differential gene expression and principal component analysis were performed using DESeq2 [32] in R (version 4.1.0). Multiple test comparisons were corrected using the Benjamini and Hochberg correction to obtain the adjusted *p*-value [33]. Genes with a log2 fold change ≥1 and an adjusted *p*-value ≤ 0.05 were considered upregulated or fissure enriched differentially expressed genes (DEGs), while those with a log2 fold change of ≤−1 and an adjusted *p*-value of ≤0.05 were considered downregulated or dorsally enriched DEGs in the analysis (i.e., fold change ≥2 or ≤0.5, respectively). Volcano plots of DEGs were drawn using the ggplot2 package in R. Enrichment analysis and functional annotation were performed in DAVID (https://david.ncifcrf.gov/home.jsp) (accessed on 1 August 2022) using *Gallus_gallus* as a background list and ENSEMBL gene IDs as inputs. Lists of GO terms were ranked and plotted (−log10(pval)) with cut-off values of *p* < 0.001.

### 2.3. In Silico Cross Species Differential Gene Expression Analysis

Sequencing data from Hardy et al., Patel et al., and Richardson et al., (References [13,14,15]) were downloaded using the SRA NCBI Toolkit. All reads were processed as follows: Reads were trimmed using Trimmomatic with parameters set as follows: LEADING:3 TRAILING:3 SLIDING WINDOW:4:15 MINLEN:40. Pairwise comparison between optic fissure and dorsal or whole eye (in chick pre-fusion samples) tissue was carried out using the Wald test in DESeq2 using raw count data generated by featureCounts. Comparisons over time were performed using the LRT test. Ensembl gene ID identifiers were converted to their mouse orthologs using the biological database network tool (https://biodbnet-abcc.ncifcrf.gov/db/dbOrtho.php). (accessed on 1 June 2022). All up-regulated genes in the fissure at any stage (absolute LFC ≥1.0, adjusted *p*-value ≤ 0.05) were selected for further analysis. Using mouse, chicken, and zebrafish up-regulated DEGs, we identified commonalities across species using the Ensembl gene database (version 102) with custom scripts based upon the Bioconductor biomaRt package [34,35] (version 4.2). Briefly, orthologs were identified using the bioDBnet biological database network (https://biodbnetbcc.ncifcrf.gov/db/dbOrtho.php). (accessed on 1 June 2022) using mouse Ensembl gene IDs as a reference. Where orthologues were not automatically identified, manual curation of the list was carried out.

### 2.4. In Situ Hybridisation Gene Expression Analyses

Chick embryos were staged according to Hamburger and Hamilton [27] and fixed overnight at 4 °C in 4 % paraformaldehyde (PFA) in 1.0 M phosphate buffered saline solution (PBS). Embryos were then rinsed in PBS twice and immersed in a 10 % sucrose-PBS solution overnight. The following day, eyes were removed and mounted in Neg-50 (Richard Allan Scientific) and snap-frozen in iso-pentane (Thermo). Sections were cut at 20 µm on a Cryostat (Leica) onto Superfrost-plus slides (Manufacturer), air dried for 2 h, and then stored at −80 °C. Slides were then thawed for 2 h at room temperature and rinsed in PBS before performing RNAscope following the manufacturer’s protocol (ACD Bio-techne, RNAscope Multiplex Fluorescent Reagent Kit v2 User Manual) and using the target-specific probes (ACD Bio-techne) detailed below in Table 1. Slides were then mounted in Fluorsave (Merck #345789) and counterstained with DAPI (ACD Bio-Techne, RNAscope Multiplex Fluorescent Reagent Kit v2 #323100). Images were captured using a Zeiss LSM 880 confocal microscope with a 20x objective and converted to JPEG file format using FIJI (ImageJ V2.1.0/1.53c) under the Creative Commons Licence.

### 2.5. Data Access at Genomics England, 100,000 Genomes Project

The 100,000 Genomes Project [36] data has been made available through the secure Genomics England (GEL) Research Embassy (RE), supported on a high-performance cluster, following information governance and security training, and with membership to the Hearing and Sight Genomics England Clinical Interpretation Partnership (GeCIP) in the rare disease programme [37]. Variant call files (VCF) generated through the Illumina Starling pipeline were passed through quality control filters as previously described (Owen et al. 2022). Participant phenotype data was present using Human Phenotype Ontology (HPO) terms [38] and accessed via a LabKey data management portal. HPO terms for anophthalmia, microphthalmia, and coloboma, as well as daughter terms, were queried using the R LabKey package. The resulting MAC cohort consisted of 346 probands.

### 2.6. Zebrafish Husbandry

Zebrafish (wild-type AB) were bred and maintained according to local UCL and UK Home Office regulations for the care and use of laboratory animals under the Animals Scientific Procedures Act at the Francis Crick Institute aquatics unit. The UCL Animal Welfare and Ethical Review Body approved all procedures for experimental protocols, in addition to the UK Home Office (License no. PPL PC916FDE7). All approved standard protocols followed the guidelines of the ARVO Statement for the Use of Animals in Ophthalmic and Vision Research Ethics.

### 2.7. CRISPR/Cas9 Mutagenesis in Zebrafish

Guide sequences targeting *emx2*, *ephb3a*, and *ephb3b* were selected using the Integrated DNA Technologies (IDT) guide RNA design tool with Refseq release 211 of the zebrafish genome. Custom Alt-R sgRNAs and SpCas9 nucleases were purchased from IDT. Zebrafish embryos were microinjected at the one-cell stage with approximately 1 nL of injection solution consisting of: 800 ng/uL Cas9 nuclease, 300 mM KCL, and 100 ng/uL of sgRNA (*emx2* only or both *ephb3a* and *ephb3b* sgRNAs). For the non-sgRNA controls, embryos were injected with all components except for sgRNA. To confirm the mutagenesis activity of the sgRNAs, DNA was extracted from 10 injected embryos at 48 hpf, and PCRs were used to amplify a region of the gene that included the target site. The amplicons were assessed by Sanger sequencing to verify that mutations were induced at the CRISPR target site. The PCR primer sequences and CRISPR guide sequences used are listed in Table 2.

### 2.8. Phenotypic Assessments

For assessing eye size, 3 and 5 dpf injected zebrafish embryos and their uninjected wild-type siblings were anaesthetized using 0.2 mg/mL tricaine and imaged using a Zeiss Stereo Discovery V20 microscope. ImageJ (FIJI) was used to measure eye diameter from the images. For each group, the mean ± standard deviation was calculated. Data were compared using either unpaired t-tests or Mann–Whitney tests. For immunofluorescence, whole zebrafish larvae at 3 dpf were fixed in 4% PFA/PBS overnight at 4 °C before washing in PBS and incubation in 30% sucrose/PBS overnight at 4 °C. The samples were mounted and frozen in TissueTek O.C.T (VWR) using dry ice. 12 μm sagittal sections were cut and collected onto Superfrost PLUS slides (Thermo Fisher Scientific, Waltham, MA, USA). After air-drying, sections were washed in PBS-0.5% Triton-X before being blocked for 1 h with 20% normal goat serum (Sigma-Aldrich) in PBS-0.5% Triton-X. and incubating with anti-laminin (Sigma-Aldrich L9393) diluted 1:50 in antibody solution (2% normal goat serum in PBS-0.5% Triton-X) at 4 °C overnight. After washing with PBS-0.5% Triton-X, the sections were incubated with Alexa Fluor 568 nm secondary antibody (Thermo Fisher) diluted 1:500 in antibody solution for 2 h at room temperature. Finally, the sections were washed and mounted in Prolong Diamond Antifade mountant + DAPI (Thermo Fisher Scientific). The slides were imaged using a Zeiss Invert 880 microscope.

## 3. Results

### 3.1. Transcriptomic Analyses during Chicken Optic Fissure Closure

We previously used dissected optic fissure tissue from chick retinas to define the transcriptional landscape during fusion [17]. However, this analysis did not include tissue from fused stages of OFC, and whole eye tissue was used as the non-fusing comparison rather than dorsal retina used in other species’ OFC transcriptomic studies [25,39]. To identify gene expression changes in the fused optic fissure and enable more closely matched cross-species comparisons, we generated transcriptomic data using RNAseq from optic fissure and dorsal retina samples collected from chicken eyes at fusing (HH St30) and the fused (HH St34) OFC stages (Figure 1a). In both cases, microdissected tissue samples were pooled depending on condition and stage (tissue from >10 embryos per condition or stage) prior to total RNA extraction, and Illumina paired end sequencing with enrichment for mRNA transcripts. Variance across the samples was assessed by principal component analysis (PCA), and these indicated that the sample replicates clustered well and that groups of samples segregated according to stage (PC1) and tissue region (PC2) (Appendix A). We then performed differential gene expression (DEG) analyses within the stages to determine fissure-specific gene expression (Figure 1b). We found 158 unique upregulated DEGs that only showed increased expression in the fusing (HH St30) fissure, 178 that were increased specifically in the fused (HH St34) fissure, and 84 genes that were enriched at both stages. Similarly, for downregulated DEGs (dorsal > fissure), there were 175 genes whose expression was reduced in the fusing fissure compared to the dorsal, 83 that were reduced in the fused fissure, and 49 that were reduced at both stages. All gene lists and ontology enrichment outputs can be found in Appendix A. We then assessed those genes with enriched expression in the fissure at each stage. In accordance with previous studies [17,25,39], at fusing stages (HH.St30), we found NTN1, ALDH1A3, VAX1, SMOC1, and PAX2 were the highest fissure-specific (Fissure > Dorsal) DEGs, whereas ALDH1A1, LYPD6, GDF6, TBX3, and TBX5 were the most dorsal-specific DEGs (Dorsal > Fissure) (Figure 1c). In silico ontology analysis of the upregulated gene set (Log2FC > 1.0, *p* < 0.05; *n* = 246 genes) in the fusing stage fissure revealed enrichment for GO terms of various developmental processes as well as those implicated in OFC, such as “ECM organisation”, “signal transduction”, “negative regulation of wound healing” and the “BMP signalling pathway”, in addition to multiple terms related to cell-cell adhesion or the regualiton of cell-junctional complexes (Figure 1d). In the fused tissue (HH.St34), NTN1, ALDH1A3, VAX1, SMOC1, and PAX2 were also identified as fissure-specific DEGs, in addition to BMPR1B, CHRDL1, and CD109 (Figure 1e). Dorsal-specific DEGs observed at both stages were TBX5, TBX3, GDF6, and LYPD6 (Figure 1c,e), consistent with previous transcriptomic analyses of OFC [17,39]. Ontology enrichment analysis for all upregulated DEGs (Log2FC > 1.0, *p* < 0.05; *n* = 262 genes) in the fused fissure (Figure 1e) revealed similar GO terms to those enriched for the fusing fissure related to cell-cell adhesion and cell-junction assembly, for example: “cell-cell adhesion via plasma-membrane adhesion molecules” and “calcium-dependent cell-cell adhesion via plasma membrane cell adhesion molecules”. Both stages are also included, “inner ear morphogenesis” and “sensory perception of sound”, likely to reflect overlap in the epithelial fusion processes involved in inner ear morphogenesis and OFC. The GO term “Axon guidance” was enriched among DEGs in both stages of OFC and includes genes involved in the ephrin, netrin, and semaphorin regulatory pathways. Interestingly, HH.St30 had fewer genes in this GO term classification (9 compared to 27) and, in contrast to HH.St34, did not include members of the UNC5 gene family, as previously shown [17,39]. In combination, these validated the use of these chicken transcriptome datasets for inclusion in subsequent analyses.

### 3.2. Cross-Species In Silico Analysis of Optic Fissure Transcriptomes

To identify evolutionarily conserved gene expression profiles in OFC, whole transcriptome RNAseq datasets from chick (this study and reference [17]), zebrafish [25], mouse and human [39] were compared. The analysis of variance using PCA analyses between species at each of the stages (pre-fusion, fusion, and fused) indicated that the samples clustered broadly according to tissue type (dorsal or fissure) and species (Figure 2a–c), although the variance for PC1 ranged from 20–38%. After alignment, differential expression analysis was performed using DESeq2 for successive pairwise condition testing (v1.18.1). Up-regulated genes were defined as those showing a log fold change (Log2FC) of >1.0 and *p* < 0.05. Analysis of upregulated genes at all stages throughout fusion among chicken, mouse, and zebrafish revealed 10 genes that were common to all three species (Figure 2d). Among these were the known human MAC genes TENM3, SMOC1, PAX2, ALDH1A3, and BMPR1B [7,26], in addition to NID1, VAX1, and NTN1, which have been previously identified as MAC or coloboma candidates based on animal studies [13,17,40]. Novel OFC genes not previously associated with MAC in either humans or animal models were EPHB3 and EMX2. We then added human transcriptome data to our analyses and found that the genes upregulated during fusion across all four species were TEMN3, ALDH1A3, NTN1, SMOC1, and VAX1 (Figure 2e). Within this group, only NTN1 and VAX1 have not previously been associated with MAC in humans. Thus, we identified EMX2 and EPHB3 as potential new OFC candidates, a core group of 10 genes whose expression was enriched in the fissure during OFC across diverse avian, mammalian, and fish species, and five genes whose expression was also enriched in the human fissure during OFC.

### 3.3. In Situ Hybridization Analysis of Enriched Gene Expression in Chicken Optic Fissures

To qualitatively confirm the transcriptome analyses and to determine if these genes have expression patterns consistent with a role in OFC or broader aspects of ventral eye development, and to identify what cells or ocular tissues these genes were expressed within, we conducted spatial in vivo gene expression analysis using fluorescence in situ hybridisation on cryosections taken from chick embryonic whole eyes that included both dorsal and ventral regions (Figure 3). Eyes were selected at stage HH28, just as fusion was beginning in the chick [17]. This approach revealed that VAX1, BMPR1B, SMOC1, ALDH1A3, NID1, and TENM3 were broadly expressed in a ventral-high, dorsal-low pattern in the developing retina and were only observed in the neural retina epithelia. Probes for SMOC1, NTN1, and ALDH1A3 gave the strongest signal, whereas EMX2, NID1, and EPHB3 were only weakly detected, broadly in line with the normalised expression values (Appendix A). NTN1 and PAX2 expression domains were restricted to the edges of the optic fissure margins, consistent with the localisation of OFC pioneer cells that mediate and directly participate in the fusion process [17,18]. ALDH1A3, BMPR1B, SMOC1, PAX2, NTN1, VAX1, TENM3, and NID1 were all expressed in the neural retina, with only some expression in the immediate RPE at the fissure edges for these genes. EMX2 expression was detected in the ventral peri-ocular regions and not detected in the RPE or neural retina. As this gene was previously reported as expressed in neuroepithelia, we used whole-mount in situ hybridisation with large (500 bp) gene-specific RNA probes to confirm the peri-ocular localisation of EMX2 expression (Appendix A). EPHB3 was broadly expressed in a range of cell types in the ventral eye. Of all these genes, the expression of only NID1 was detected in the dorsal retina. Trajectory analyses of expression levels relative to dorsal tissue throughout OFC progression (pre-fusion, fusing, and fused) revealed that no clear temporal gene expression patterns emerged, except that ALDH1A3 and EMX2 both decreased after fusion for all species. BMPR1B, SMOC1, and TENM3 either increased expression in the fissure or it remained constant or slightly decreased after fusion, whereas NID1, NTN1, VAX2, PAX2, and EPHB3 all showed decreased expression levels after fusion in 3/4 species (Appendix A).

### 3.4. Coloboma Candidate Gene Targeting In Vivo

To determine the requirement of the novel genes identified in this study for eye development and OFC, we generated F0 knock-out zebrafish embryos using CRISPR/Cas9 gene editing (GE) targeting exon DNA immediately downstream of the translation initiation site (ATG) and analysed their eye developmental progress at two stages (Figure 4). The zebrafish genome has two duplicated ephb3 genes, ephb3ba and ephb3bb, and therefore we designed guide RNAs for both loci to ensure we generated complete removal of these gene products whose activities could be functionally redundant. Both emx2 and ephb3 loss-of-function gene editing events were confirmed by PCR from genomic DNA (Figure 4a). In zebrafish, OFC is complete by 56 h post fertilisation [41], and compared to wild-type (not injected) and Cas9-only injected embryos, the GE embryos had developed normally at 3 days post fertilisation (dpf) and 5 dpf (Figure 4b), although eye sizes were apparently reduced. Measurements of eye diameters revealed smaller eye size in both GE groups compared to controls at 3 dpf and 5 dpf (Figure 4b,c). Macroscopic analysis revealed coloboma phenotypes in ephb3ab (47%, *n* = 15) and emx2 (27%, *n* = 15) targeted embryos. In addition, sections cut from control and GE eyes and stained for laminin showed persistence of the basement membrane surrounding the edges of the optic fissure, consistent with fusion failure and coloboma in these embryos (Figure 4d). Thus, of the 10 genes identified in this study whose expression in the ventral retina during OFC is conserved across different vertebrates, all can be associated with coloboma or microphthalmia causation in animals or humans when their function is perturbed and can be considered as MAC candidates for diagnostic screens.

### 3.5. Screening MAC Cohorts for Human Variants in Novel OFC Genes

We then sought to identify any novel disease-causing variants associated with MAC in EMX2 and EPHB3. We also included NID1 on the basis of our data and the phenotype previously observed in nid1-targeted zebrafish [40]. A survey of the whole genome sequencing (GS) data of genetically unsolved MAC probands within the Genomics England 100,000 Genomes Project were interrogated for variants either assigned as Tier III or variants of uncertain significance (VUS) as per ACMG-AMP guidelines involving the candidate genes. Sequencing data provided a minimum coverage of 15 times for more than 97% of the autosomal genome and were subsequently aligned to either GRCh37 or GRCh38 of the human genome (Isaac, Illumina Inc., San Diego, CA, USA). Variants were generated, including single-nucleotide variants and indels (insertions or deletions) (Starling, Illumina Inc.), copy number variants (CNV, Canvas, Illumina Inc.), and structural variants (SV, Manta, Illumina Inc.). Genomic coordinates of candidate genes were identified and used to positionally filter proband variant files, followed by further annotation using dbNSFP (v4.3a). Variants were prioritised in a stepwise manner by filtering using minor allele frequency in publicly available and in-house datasets, predicted functional protein consequences, and familial segregation. Surviving variants were further inspected visually using IGV (Table 3).

Screening of the proband cohort identified a heterozygous VUS within EPHB3, EPHB3(NM_004443.4):c.649G>A,(p.Ala217Thr) for which several in silico predictions reported a damaging consequence, although some were benign (CADD 24.7, FATHMM pathogenic, Polyphen2 damaging, and EIGEN pathogenic). The gnomAD genome allele frequency (v3.1.2) for this variant (GRCh38:3-184572969-G-A) reports a total of 11/152182 alleles, although 10/5184 alleles are observed in the East Asian population, the proband ethnicity was White. No segregation analysis was undertaken as the sample was a singleton. We identified a second heterozygous VUS, EPHB3(NM_004443.4):c.1259G>A (p.Arg420His) in a male proband with HPO in terms of autism, microphthalmia, seizures, and true anophthalmia. Whilst the population frequency reported was low, in silico prediction of functional consequence was indeterminate.

Variants within the NID1 gene were also prioritised; however, no variants survived our filtering. We report a heterozygous variant NID1 (NM_002508.3):c.2204G>A (p.Arg735His), which has been previously identified as pathogenic in a patient with neural tube defects [42]; however, the minor allele frequency was too high to enable us to term this as a variant of interest, although it may warrant further investigation (note that this has been reported here for completeness). We also report two additional NID1 coding variants: c.3458C>G(p.Pro1153Arg) and c.2859G>T(p.Lys953Asn) for which in silico predictions for consequence to protein range from uncertain to pathogenic. These alleles were not found in gnomAD. No copy number or structural variants were identified in any of the candidate genes screened. Amino acid alignments of the protein sequence adjacent to the varaints are shown for all four species in Appendix A.

## 4. Discussion

The main purpose of this study was to identify novel plausible MAC candidates, and in this endeavour, we found two new genes, *EMX2* and *EPHB3*, that should now be associated with eye development and considered plausible disease genes for optic fissure closure defects. Our cross-species meta-analysis also provided evidence that genes previously associated with coloboma causation in model organisms, such as *VAX1*, *NTN1*, and *NID1*, should be more widely recognised as possible human MAC candidates.

The identification of novel MAC genes is challenging, largely due to the poor recurrence rates among non-related affected families, genetic heterogeneity, limits in sample sizes, and human phenotype ontology inconsistencies [21,23]. Single variants, even if they are predicted to be damaging or are shown in animal models as deleterious and phenocopy the human phenotype, still do not meet the strict criteria to be considered causative [43]. Here, we present novel variants and report new genes associated with MAC; however, we were unable to unambiguously confirm their causality. Microphthalmic eyes may occur with colobomas, as the failure of the optic fissures to physically become apposed in undersized eye cups during development inevitably prevents fusion from occurring. However, in some cases, colobomas can occur in normal-sized eyes, indicating that in these colobomas the pathogenesis results from a direct failure in the fusion processes rather than from eye cup growth. Identifying coloboma candidate genes and separating these from microphthalmia genes is therefore challenging, largely due to the difficulties in isolating the cells that govern tissue fusion at the correct time during development [16]. Indeed, this is a highly complex process involving small populations of cells from both the edges of the retinal epithelium and the peri-ocular mesenchyme (POM), that directly or indirectly mediate fusion in the eye, respectively [17,18,44]. We hypothesised that performing in silico data mining of vertebrate OFC transcriptomes may yield novel genes that are expressed in the pioneer cells or in the adjacent POM. This suffered from the inherent limitations that OFC in these diverse vertebrates is not identical, for example, in the timing and duration of the OFC process and the overall size of the eye [16]. In addition, there are inter-species differences in the presence or absence of apoptotic foci in pioneer cells [17,39] and the intercalation of optic nerve cells and astrocytes at the proximal but not medial region of the fissure [45]. We aimed to overcome this second difference by carefully dissecting the medial and not the proximal OF from our chicken retinas. Despite these differences, and similar to a previous study [26], we successfully showed conservation of gene expression across these species during OFC and found that the majority of these genes had already been associated with MAC phenotypes, further validating our approach. Given the broad ventral expression domains of the majority of genes we identified, these are unlikely to directly regulate fusion processes and are instead considered more likely to be associated with microphthalmia or a combination of microphthalmia and coloboma.

We identified POM-specific expression of *EMX2* during OFC and showed that its targeted disruption led to MAC phenotypes in zebrafish; however, we were unable to identify any novel pioneer cell markers. We propose that the identity of pioneer cells is highly transient, and due to the rapid changes in cell behaviour required for fusion, their unique molecular signatures may only be revealed through single-cell transcriptomic analyses rather than through bulk RNAseq approaches. However, the generation of a pioneer cell reporter line in any of these species could enable the selective isolation of these cells and their subsequent characterization throughout the OFC process at multiple omics levels. Our study shows that the gene expression conservation we observed in OFC among different vertebrates would permit either of these model species and approaches to be used for such experiments, and we propose that these will be the logical next steps for OFC research and OC candidate gene identification.

EMX2 is a homeobox transcription factor required for central nervous system and urogenital development [46,47]. It is a homologue of the *Drosophila empty spiracles* (*ems*) gene that is essential for anterior head development and brain segmentation. Mice lacking functional Emx2 die at birth and have no kidneys or reproductive organs [46]. In humans, heterozygous *EMX2* mutations have been associated with Schizencephaly [OMIM #269160], but no MAC or other ocular malformations have previously been reported associated with mutations in this gene. Our study provides the first example of genetic targeting of *EMX2* to cause developmental eye disorders, with microphthalmia and coloboma observed in Crispr/Cas9-generated mutant *emx2* zebrafish. On review of the available literature, an apparent coloboma and ventral eye defect in a single image panel of an *Emx2*^-/-^ mouse embryo at gestational stage E12.5 was noted, when OFC should be complete [48]. As this was not reported by the authors, without further data, it is unclear what the penetrance or underpinning mechanism of this phenotype is; however, this appears to support our data indicating the necessity for EMX2 in eye development. Further investigation is required to understand its precise role in OFC.

*NID1* encodes nidogen, a component of the basement membrane that overlies the edges of the fissures as they become apposed prior to fusion. A previous study [40] found that the zebrafish orthologue *nid1* is one of several paralogous nidogens expressed in the fissure margin, but whose expression is downregulated specifically at the onset of fusion, presumably helping to mediate the remodelling of the basement membrane (BM) to enable fusion through either reducing the integrity of the BM or by exposing proteolytic sites in the remaining components of the OFM BM. We found *NID1* among the 3-species list (chickens, zebrafish, and mice) of OFM genes and found that its expression was reduced in fused versus fusing transcriptomes. We also showed that its expression was localised to the apical-most cells of the chicken ventral neural retina. Morpholino-based knock-down of *nid1* in the previously mentioned study caused coloboma [40], and therefore we did not generate mutants for this gene. However, given the zebrafish phenotype and the conservation of OFM *NID1* expression across the range of species shown here, we suggest this gene should be considered in the genetic analysis of MAC patients.

Ephrin type-B receptor 3 is a transmembrane tyrosine kinase protein with affinity for the ephrin-B family of ligands and is encoded by the *EPHB3* gene. Ephrin receptor-ligand interactions at cell surfaces mediate numerous dynamic developmental processes, such as migration, proliferation, and cell fate determination, mediated through changes to cytoskeletal dynamics [49], and can also act as tumour suppressor genes [50]. A role for Ephrin signaling in eye development is also emerging: dominant mutations in *EPHA2* cause isolated congenital cataracts, and its ligand *EFNA5* is important for normal lens development [51,52,53,54]. Heterozygous pathogenic *EPHA2* variants have also been reported in two families presenting with congenital cataract and non-syndromic microphthalmia, widening the phenotype associated with this gene [55]. Even more recently, biallelic mutations in *EPHA2* have been found in a syndromic form of microphthalmia with anterior segment dysgenesis [56]. The variants identified in this study cannot, at this point, be considered pathogenic without further functional analyses. Mutations in *EPHB3* have not yet been associated with defects in eye development, but the zebrafish CRISPR/Cas9 mutant phenotype data presented here together with its developmental expression in the ventral chicken eye tissue suggests this gene has an important role in normal oculogenesis and OFC and should warrant further study in this context.

The ventral vertebrate retina remains undifferentiated during optic cup development largely through its unique transcriptional signature. The gene expression data presented in this study suggests this is largely conserved, with *VAX1*, *ALDH1A3*, *SMOC1*, *NTN1*, and *TENM3* all detected as ventrally expressed genes across human, mouse, chicken, and zebrafish species. Interestingly, *PAX2*, which when mutated is known to cause ocular coloboma in humans and zebrafish [57,58], was upregulated in mouse, chicken, and zebrafish fissures and was highly specific to the pioneer cell region of the retina. However, *PAX2* was not included as a conserved OFC-enriched gene when we added human expression data. This is likely due to the limitations of the datasets used within our in silico approach, as the human dataset from Patel et al. [39] showed variance in expression levels between dorsal and fissure tissues across the human samples analysed (*n* = 3). Furthermore, *PAX2* was not featured in their list of significantly differentially expressed genes from the human and mouse fissure margins [39]. This highlights the difficulty in obtaining high-quality human samples and reinforces the utility of multi-species approaches for OFC research.

The relative frequency of colobomas in the gene-edited zebrafish was suggestive of incomplete penetrance and possible modifier effects from other genetic loci or indicated redundancy from paralogous proteins. Although we did not formally test for off-target edits in other genetic loci that may contribute to these phenotypes, these seem unlikely from the in silico off-target predictions determined by the guide RNA generation software. Nevertheless, further work will be required with breeding these stable lines to enable in-depth mechanistic analyses of the gene function in ventral eye development.

In summary, we applied an in-silico pipeline combining new and existing RNAseq datasets to reveal genes whose expression is enriched in the ventral eye during optic fissure closure stages of development across a range of diverse species. This approach was coupled to in vivo validation by spatial analysis of gene expression and revealed a core group of genes that are expressed during OFC; five were already known human MAC genes (*ALDH1A3*, *BMPR1B*, *PAX2*, *TENM3*, and *SMOC1*), and three (*NID1*, *NTN1*, and *VAX1*) were previously associated with structural eye defects in animal models. All our conserved OFC genes, including the two novel MAC candidates, *EMX2* and *EPHB3*, should be considered for further genetic and molecular analyses based on their gene expression profiles during eye development and the presence of coloboma and microphthalmia phenotypes in animal models. This study expands the use of developmental expression analyses in model systems to provide supportive evidence towards identifying candidate genes for human structural eye malformations.

## Figures and Tables

**Figure 1 biomolecules-13-00293-f001:**
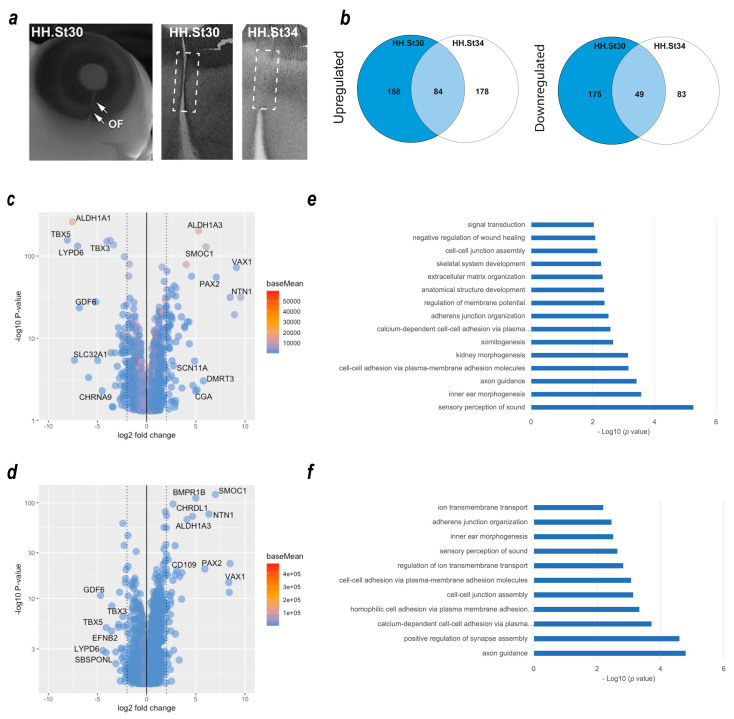
Transcriptome profiling of chicken optic fissure closure. (**a**) Schema for segmental dissection of chick OFM tissue at the fusing (HH.St30) and fused (HH.St34) OFC stages, with arrows indicating the limits of the optic fissure margin. (**b**) Venn diagrams representing the number of DEGs that were (left) significantly upregulated: Fissure expression > Dorsal, or (right) downregulated: Dorsal expression > fissure) between the fusing HH.st30 fissure (blue circle) and fused HH.st34 fissure tissue (clear circle). The number of DEGs that are commonly upregulated/downregulated between the two stages is indicated in the overlapping region. (**c**) Volcano plot showing differentially expressed genes in the fusing stage HH.st30 OFC transcriptomes. (**d**) Volcano plot showing differentially expressed genes in the fused stage HH.st34 transcriptomes. Genes with the greatest differential changes are labelled for each graph. (**e**,**f**) Plots showing DAVID enrichment analysis outputs for fissure-specific DEGs at the two developmental stages ((**e**), HH.St30; (**f**), HH.St34). Values shown are −Log10 (*p* value). All accompanying enrichment and gene expression analysis data are in Appendix A.

**Figure 2 biomolecules-13-00293-f002:**
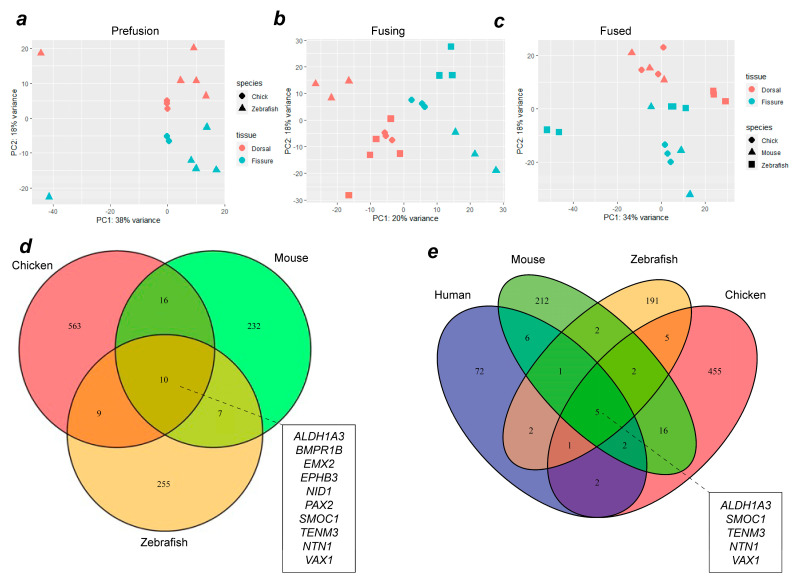
Combined transcriptome analysis across vertebrate species reveals common gene expression signatures during OFC. PCA analyses for aligned transcriptomes for chicks and zebrafish (**a**), or for chicks, zebrafish, and mice (**b**,**c**) at OFC stages prior to fusion (**a**), active fusion (**b**), or fused fissure stages (**c**). (**d**) Venn diagram showing intersection analysis for fissure-enriched DEGs for all stages of OFC between mice, chicks, and zebrafish. There were 10 fissure-specific DEGs shared among all three species: EPHB3, ALDH1A3, BMPR1B, EMX2, NTN1, PAX2, SMOC1, VAX1, TENM3, and NID1. (**e**) Analysis of shared DEGs among all four species identified five genes with fissure-enriched expression throughout OFC in humans, mice, chickens, and zebrafish: SMOC1, NTN1, ALDH1A3, VAX1, and TENM3. Additional genes whose expression is conserved to humans from each single species are listed in Appendix A.

**Figure 3 biomolecules-13-00293-f003:**
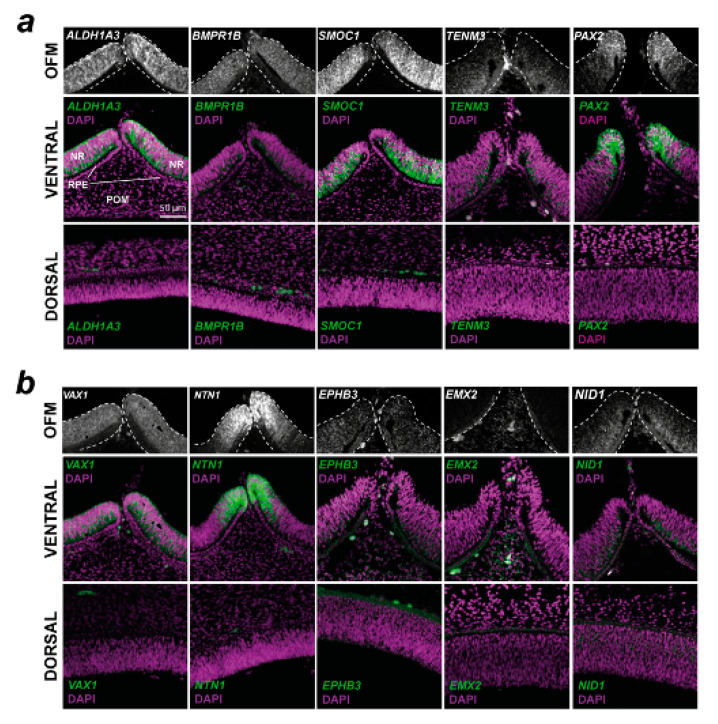
Spatial gene expression analysis for 10 conserved fissure-enriched genes in the developing chick eye. Fluorescent in situ hybridisation for (**a**) known coloboma gene targets and (**b**) OFC candidate genes identified in this study was counterstained with DAPI nuclear stain on chick eyes at HH28, immediately prior to fusion. Note that for EMX2, the OFM region highlights the POM region; expression was not observed in the NR or RPE. In both (**a**,**b**), Top: grayscale RNAscope probe signal without DAPI highlighting the mRNA signal in the immediate optic fissure margin region. Middle: mRNA probe and DAPI merged images of the broad ventral eye region. Bottom: Dorsal eye region with probe and DAPI merged. OFM, optic fissure margin; NR, neural retina; POM, periocular mesenchyme; RPE, retinal pigmented epithelium. Scale = 50 µm.

**Figure 4 biomolecules-13-00293-f004:**
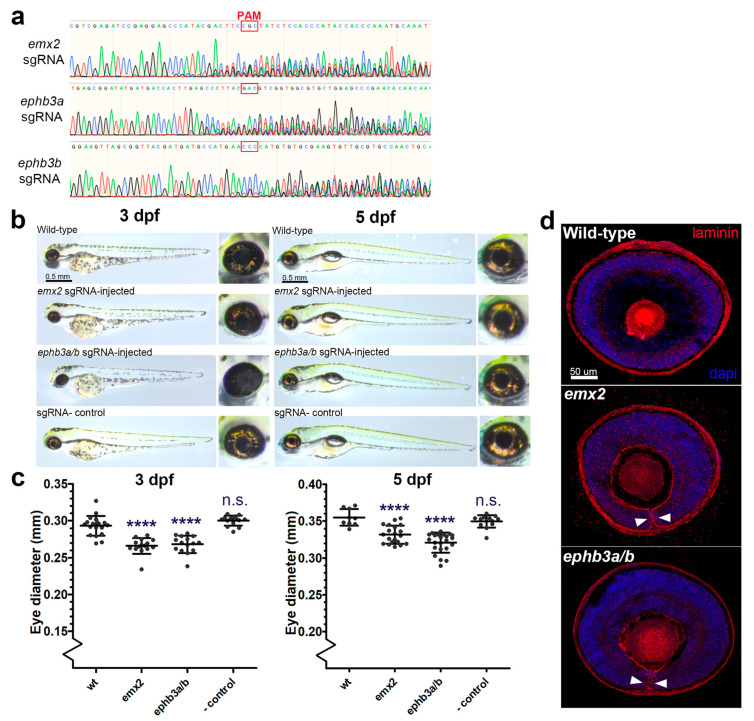
Targeted gene disruption of OFC novel candidates using CRISPR/Cas9 gene targeting in zebrafish embryos. (**a**) Sanger sequence traces for genomic DNA PCR analysis of CRISPR/Cas9 targeted embryos show successful genetic disruption at the sgRNA site loci for eph3ba, eph3bb, and emx2. PAM sites are indicated. (**b**) Brightfield microscopy analysis of targeted fish at 3 dpf and 5 dpf compared to uninjected controls and Cas9-only controls. (**c**) Eye diameter metrics for fish injected with CRISPR/Cas9 at 3 dpf and 5 dpf (**** = *p* < 0.0001). (**d**) Confocal analysis of laminin-stained cryosections from gene-edited and control eyes showing persistence of the basement membrane in the optic fissure margin (arrowheads). Colobomas were observed in 27% and 47% of gene-edited embryos analysed for *emx2* and *ephb3a/b*, respectively (*n* = 15 per target gene).

**Table 1 biomolecules-13-00293-t001:** RNAscope Custom-designed and Catalogue Probes Used.

mRNA Target	Name of Probe	Channel	Catalogue Number
*TENM3*	*Gg-TENM3-C2*	C2	1147351-C2
*EMX2* (Transcript variant X1)	*Gg-EMX2-C2*	C2	1060951-C2
*PAX2*	*Gg-PAX2-C2*	C2	1147371-C2
*EPHB3* (Transcript variant X7)	*Gg-EPHB3-C2*	C2	458861-C2
*NID1*	*Gg-NID1-C1*	C1	1147381-C1
*ALDH1A3*	*Gg-ALDH1A3-C2*	C2	1144151-C2
*BMPR1B*	*Gg-BMPR1B-C1*	C1	1144161-C1
*VAX1*	*Gg-VAX1-C1*	C1	1144181-C1
*SMOC1* (Transcript variant X4)	*Gg-SMOC1-C2*	C2	593601-C2
*NTN1*	*Gg-NTN1-C2*	C2	497491-C2

**Table 2 biomolecules-13-00293-t002:** Oligonucleotide sequences used for zebrafish gene editing and genotyping.

Gene	CRISPR Guide Sequence (5′−3′)	Primer Sequences (5′−3′)
*emx2*	CGAGGAGCCCATACGACCAG	Forward: CACGATGTGTTGAGCTGTGC Reverse: CCTTTGCTGGCTTGCGAAAA
*ephb3a*	ATGACCACTTGAGCCCCATC	Forward: TTACATTCCACCTGCTTACACC Reverse: ACATAAGGATTCTCCCTCCACG
*ephb3b*	ACACTTGGTACGTTCGGATG	Forward: GCAGTACCTTTGCAGCGTAAC Reverse: GAAGCTCTCATCCGGAGCAA

**Table 3 biomolecules-13-00293-t003:** Summary of in silico analysis of single nucleotide variants (SNVs) identified. Screening of probands enrolled in the 100,000 Genomes Project presenting with HPO terms including microphthalmia, anophthalmia, or coloboma for in silico predicted damaging variants resulted in five families. The cDNA position of the SNV, the subsequent amino acid change, and the genomic location for the human reference genome GRCh38 are presented. All the variants identified were heterozygous. Outcomes of the predictive algorithms SIFT, Polyphen-2 and MutationTaster, REVEL, and UCSC PhyloP100 are reported, with respective scaled CADD scores and allele frequency as reported in gnomAD (https://gnomad.broadinstitute.org, accessed on 1 October 2022, NF—not found). For NID1, no Dandy-Walker phenotypes were identified in the HPO terms for these individuals.

Family	HPO Terms	Gene	cDNA/GRCh38	Protein	SIFT	Polyphen-2	Mutation Taster	REVEL	PhyloP100	CADD	gnomAD
F1	Optic nerve coloboma, moderate proteinuria, stage 3 chronic kidney disease	*EPHB3*	NM_004443.4 c.649 G>A 3:184572969:G:T	p.(Ala217Thr)	Tolerated	Probably Damaging	Disease Causing	0.27	7.965	24.7	0.001%
F2	Intellectual disability, autism, microphthalmia, seizures, true anophthalmia	*EPHB3*	NM_004443.4 c.1259 G>A 3:184577088:G:A	p.(Arg420His)	Benign	Probably Damaging	Disease Causing	0.069	3.343	22.6	3 × 10^−5^
F3	Chorioretinal coloboma, macrocephaly, unilateral cleft lip, sleep apnea, abnormal aggressive impulsive or violent behaviour	*NID1*	NM_002508.3 c.2204 G>A 1:236017198:C:T	p.(Arg735His)	Deleterious	Possibly Damaging	Disease Causing	0.575	7.456	24.04	0.016
F4	Bilateral retinal coloboma, mild global developmental delay, abnormality of the vertebrae, bulbar palsy, Sprengel anomaly, abnormality of the ear, skeletal system, nervous system	*NID1*	NM_002508.3 c.3458 C>G 1:235979873:G:C	p.(Pro1153Arg)	Pathogenic	Probably Damaging	Uncertain	0.671	10.003	25.9	NF
F5	Retinal coloboma, anterior segment dysgenesis, microphthalmia, chorioretinal coloboma, iris coloboma	*NID1*	NM_002508.3 c.2859 G>T 1:235990955:C:A	p.(Lys953Asn)	Uncertain	Probably Damaging	Uncertain	0.192	1.649	24.4	NF

## Data Availability

All chicken RNAseq data files generated for this study are submitted to the NCBI Gene Expression Omnibus database (http://www.ncbi.nlm.nih.gov/geo) with accession number GSE224379.

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
