# Peer review of "Identification of Novel Coloboma Candidate Genes through Conserved Gene Expression Analyses across Four Vertebrate Species"

_biomolecules, 2023, doi:10.3390/biom13020293_

Round 1

Reviewer 1 Report

The authors presented a very convincing exploratory study on the potential genes involved in the optic fissure closure. The integration of different gene expression studies is an important strength of the study. The creative usage of past and new datasets as well as relevant bioinformatics tools make the identified genes stand out as relevant genes involved in OFC and potentially relevant clinical screening targets.

Minor comments:

1. Page 1, line 28-29: "scrutinized" in the correct word.

2. Page 5, table 1: '*' usage is confusing. Footnote is more appropriate.

3. Page 8, line 221: Please clarify what "pan-specific comparisons" are or simplify the sentence.

4. Page 8: line 223: "tableure 1a" needs to be replaced by an English term.

5. Please discuss whether you have some data on whether many relevant OFC genes get lost during the cross-species data intersection and whether two-species comparisons or modeling human variants in e.g. zebrafish could be more sensitive approaches.

6. Please discuss to what extent OFC and microphthalmia are functionally related in the context of this study and in general.

Author Response

Reviewer 1.

The authors presented a very convincing exploratory study on the potential genes involved in the optic fissure closure. The integration of different gene expression studies is an important strength of the study. The creative usage of past and new datasets as well as relevant bioinformatics tools make the identified genes stand out as relevant genes involved in OFC and potentially relevant clinical screening targets.

Minor comments:

  • Page 1, line 28-29: "scrutinized" in the correct word.

We have changed this passage to:

"We analysed available whole genome sequencing datasets from MAC patients and identified a range of variants with plausible causality."

1.2. Page 5, table 1: '*' usage is confusing. Footnote is more appropriate.

We have removed the asterisk as it is now redundant, these probes are all now available through the manufacturer’s catalogue.

  • Page 8, line 221: Please clarify what "pan-specific comparisons" are or simplify the sentence.

We have changed this passage in the text to:

"To identify gene expression changes in the fused optic fissure and enable more closely matched cross species comparisons, we generated transcriptomic data using RNAseq from optic fissure and dorsal retina samples collected from chicken eyes at fusing (HH St30) and fused (HH St34) OFC stages (Figure 1a)."

  • Page 8: line 223: "tableure 1a" needs to be replaced by an English term.

We were unable to find this typographical error but have searched and ensured this term does not exist in the text.

  • Please discuss whether you have some data on whether many relevant OFC genes get lost during the cross-species data intersection and whether two-species comparisons or modeling human variants in e.g. zebrafish could be more sensitive approaches.

We thank the reviewer for this interesting question. We had previously performed analyses that included paired analyses across all species. We found there were no known MAC genes present in these that were subsequently lost in the 3x or 4x species analyses. Those genes that were not known MAC genes that were lost can still be found in any of the OFM gene lists in existing publications that have performed transcriptional profiling in these species (Hardy et al., 2019; Richardson et al., 2019; Patel et al., 2020). Thus, we did not feel adding these lists provided any additional value in this paper.

1.6. Please discuss to what extent OFC and microphthalmia are functionally related in the context of this study and in general.

We have referred to this in the main introduction and discussion of the original manuscript, especially focusing on the process of fusion itself. However, to make more direct reference to this, we have added some more information to the discussion (para 2, line 512-520):

"Microphthalmic eyes may occur with colobomas, as the failure of the optic fissures to physically become apposed in undersized eye cups during development inevitably prevents or disrupts fusion from occurring. However, in some cases colobomas can occur in normal sized eyes, indicating that in these colobomas the pathogenesis results as a direct failure in the fusion processes, rather than eye cup growth. Identifying coloboma candidate genes and separating these from microphthalmia genes is therefore challenging, largely due to the difficulties in isolating the cells that govern tissue fusion itself [16]. This is a highly complex process, involving small populations of cells from both the edges of the retinal epithelium, and the peri-ocular mesenchyme (POM) that directly, or indirectly, mediate fusion in the eye, respectively [17,18,44]."

We have also added this sentence (page 20, line 540-543):

"Given the broad ventral expression domains of the majority of genes we identified, these are unlikely to directly regulate fusion processes and are instead considered more likely to be associated with microphthalmia, or microphthalmia and coloboma combined."

Reviewer 2 Report

Trejo-Reveles et al. detail the identification of novel coloboma candidate genes by examining gene expression analyses across four vertebrate species.  This is an important study much of the genetic variation that causes coloboma is unknown with only ~20-33% of disease explained, and this is a devastating cause of childhood blindness.  The manuscript potentially implicates three new genes EMX2, NID1, and EPHB3 with in the pathogenesis of microphthalmia/coloboma.  For one of these genes moderate quality genetic evidence is provided to support their role in human MAC, but for the remaining genes, the genetic data is less convincing.  As such, some of the paper’s conclusions and writing need to be modified to reflect this, unless additional data can be provided to support the role of this gene (EPHB3) and new variants found for EMX2.  These would be candidate genes for research panels, but should not be included in clinical diagnostic testing panels as of yet as they would be GUS/VUS at best.  Overall, the manuscript is well-written and impactful, but the following need to be addressed prior to publication:

1)       “In combination our data suggest that expression of genes involved in ventral eye development are conserved across a range of vertebrate species, and that EMX2, NID1, and EPHB3 are candidate loci that should be adopted into clinical diagnostic screens for patients with structural eye malformations.”

This comment is premature, as the data presented in this manuscript support the role of these as candidate genes, but do not provide definitive association with disease and thus would only be actionable in a research and not a clinical context. 

2)       For Venn diagram in Figure 2 – DEGs between human and 1 of the other species may also be important for disease pathogenesis in humans, so would recommend a supplemental table that details the DEGs of each species and overlap with human (the remaining 14 genes that are DEGs in human and at least one species), not just highlighting the 5 genes that are conserved DEGs across all 4 species.

3)       Figure 4 – the scaled graphs are misleading, please start at 0 mm for each of the scales, as the eye size differences are quite small (albeit convincingly statistically significant). 

4)       The authors provide quantitation of eye size, but not the fraction of eyes that developed coloboma.  This is an important finding to support the robustness of the zebrafish CRISPR editing.  Would recommend the fraction of the eyes that develop coloboma be quantified for each of the emx2 and ephb3a/b sgRNA experiments as this would support a role for these genes in MAC.  If it is an infrequent phenotype, CRISPR offtarget effects would need to be ruled out or discussed – either by breeding the fish to show stable transmission of the eye size coloboma phenotypes or by other genomic sequencing of the zebrafish to rule out targeting of other coloboma genes. 

5)       Rationale should be provided for screening cohorts for EMX2, EPHB3, NID1, as NID1 comes a bit out of the context in this part of the results (section 3.5 line 387).  It has been associated with MAC in animals, but not in human disease.  A sentence describing the rationale should be sufficient. 

6)       As noted above, the genetic evidence for EPHB3 as a human disease gene are not sufficiently strong to suggest inclusion in panels and not entirely convincing.  While the data support its role in fissure closure and eye size in zebrafish and its expression, the two variants found in patients are not that compelling without additional functional evidence.  The p.(Ala217Thr) variant is at a high population frequency in East Asians, beyond the population frequency at which MAC would be expected to be observed in a healthy population of controls (even with some variability in penetrance) – if the authors disagree would need to provide examples of MAC alleles with gnomad population allele frequency in this range.  The other variant has conflicting in silico predictions and is not conserved in zebrafish or chicken, and has a relatively low REVEL score – 0.069.   Another metapredictor may be helpful, i.e. REVEL to include for Table 3.  To strength their argument of EPHB3 as a disease gene, either additional structural analysis (AlphaFold predictions) or functional analysis – generation of Ala217Thr orthologous mutation in zebrafish would be necessary to help establish pathogenicity of these alleles.  Otherwise, the association should be downplayed only a “possible” one.

7)       Genetic data for NID1 is more convincing with the exception of the Arg735His variant.  This one should be removed and placed in supplemental information.  Based on the discussion in the manuscript, the authors included it for completeness and generally agree with this assessment.    There is a high gnomad allele frequency and 36 homozygotes in gnomad suggesting this is likely to not cause such a severe phenotype in the patient.  For NID1 patients, since this has been associated with Dandy-Walker malformations a comment on whether these individuals were phenotyped for this malformation would be helpful.  As noted above, REVEL and molecular modeling may be helpful for further strengthening in silico pathogenicity. 

8)       The discussion is lacking in discussing the EPHB3 variants and potential pathogenicity

9)       Given this is an evolutionary conservation based study, it would be interesting for the authors to report conservation of variants they observed in patients among the 4 species studied. 

10)   Line 236 “expresison” should be expression

Author Response

Trejo-Reveles et al. detail the identification of novel coloboma candidate genes by examining gene expression analyses across four vertebrate species.  This is an important study much of the genetic variation that causes coloboma is unknown with only ~20-33% of disease explained, and this is a devastating cause of childhood blindness.  The manuscript potentially implicates three new genes EMX2, NID1, and EPHB3 with in the pathogenesis of microphthalmia/coloboma.  For one of these genes moderate quality genetic evidence is provided to support their role in human MAC, but for the remaining genes, the genetic data is less convincing.  As such, some of the paper’s conclusions and writing need to be modified to reflect this, unless additional data can be provided to support the role of this gene (EPHB3) and new variants found for EMX2.  These would be candidate genes for research panels, but should not be included in clinical diagnostic testing panels as of yet as they would be GUS/VUS at best.  Overall, the manuscript is well-written and impactful, but the following need to be addressed prior to publication:

2.1       “In combination our data suggest that expression of genes involved in ventral eye development are conserved across a range of vertebrate species, and that EMX2, NID1, and EPHB3 are candidate loci that should be adopted into clinical diagnostic screens for patients with structural eye malformations.” This comment is premature, as the data presented in this manuscript support the role of these as candidate genes, but do not provide definitive association with disease and thus would only be actionable in a research and not a clinical context. 

We thank the reviewers for this wise comment and in response we have changed the text to:

“.. EMX2, NID1, and EPHB3 are candidate loci that warrant further functional analysis in the context of MAC, and should be considered for sequencing in cohorts of patients with structural eye malformations.”

2.2       For Venn diagram in Figure 2 – DEGs between human and 1 of the other species may also be important for disease pathogenesis in humans, so would recommend a supplemental table that details the DEGs of each species and overlap with human (the remaining 14 genes that are DEGs in human and at least one species), not just highlighting the 5 genes that are conserved DEGs across all 4 species.

We thank the reviewer for this suggestion and have added these into a Supplemental Table (Table S2).

2.3)       Figure 4 – the scaled graphs are misleading, please start at 0 mm for each of the scales, as the eye size differences are quite small (albeit convincingly statistically significant). 

We apologise that this graph looked misleading. We have adjusted the Y axis scale on the graphs, however we chose not to start at 0 mm as requested by the reviewer, as the values would be unreadable. Instead, we have added a clear break in the axis  to show that there is a jump in the scale and make it less misleading.

2.4)       The authors provide quantitation of eye size, but not the fraction of eyes that developed coloboma.  This is an important finding to support the robustness of the zebrafish CRISPR editing.  Would recommend the fraction of the eyes that develop coloboma be quantified for each of the emx2 and ephb3a/b sgRNA experiments as this would support a role for these genes in MAC.  If it is an infrequent phenotype, CRISPR off target effects would need to be ruled out or discussed – either by breeding the fish to show stable transmission of the eye size coloboma phenotypes or by other genomic sequencing of the zebrafish to rule out targeting of other coloboma genes. 

For the gene editing zebrafish coloboma data - we determined the proportion of emx2 and ephb3a/b embryos that have a coloboma at 3dpf to be 27% and 47%, respectively (WT= 0%). We have added these data to Figure 4 legend, and into the main text in the results section (P16, line 378-9). We have also added a short discussion point (P23, lines 557-564) we could add that we can't rule out the possibility of off-target effects of CRISPR and that it would be worth investigating stable lines in the future:

"The relative frequency of colobomas in the gene-edited zebrafish were suggestive of incomplete penetrance and possible modifier effects from other genetic loci, or indicate redundancy from paralogous proteins. Although we did not formally test for off-target edits in other genetic loci that may contribute to these phenotypes, these seem unlikely from the in silico off-target predictions determined by the guide RNA generation software. Nevertheless, further work will be required with breeding these stable lines to enable in-depth mechanistic analyses of these genes function in ventral eye development."

2.5)       Rationale should be provided for screening cohorts for EMX2, EPHB3, NID1, as NID1 comes a bit out of the context in this part of the results (section 3.5 line 387).  It has been associated with MAC in animals, but not in human disease

We agree that a short justification for the inclusion of NID1 is required and have modified the main text accordingly:

"We then sought to identify any novel disease-causing variants associated with MAC in EMX2 and EPHB3. We also included NID1 on the basis of our data and the phenotype previously observed in nid1 targeted zebrafish [40]."

2.6)       As noted above, the genetic evidence for EPHB3 as a human disease gene are not sufficiently strong to suggest inclusion in panels and not entirely convincing.  While the data support its role in fissure closure and eye size in zebrafish and its expression, the two variants found in patients are not that compelling without additional functional evidence.  The p.(Ala217Thr) variant is at a high population frequency in East Asians, beyond the population frequency at which MAC would be expected to be observed in a healthy population of controls (even with some variability in penetrance) – if the authors disagree would need to provide examples of MAC alleles with gnomad population allele frequency in this range.  The other variant has conflicting in silico predictions and is not conserved in zebrafish or chicken, and has a relatively low REVEL score – 0.069.   Another metapredictor may be helpful, i.e. REVEL to include for Table 3.  To strength their argument of EPHB3 as a disease gene, either additional structural analysis (AlphaFold predictions) or functional analysis – generation of Ala217Thr orthologous mutation in zebrafish would be necessary to help establish pathogenicity of these alleles.  Otherwise, the association should be downplayed only a “possible” one.

We thank the reviewer for these important discussion points. We agree that the human genetic data is currently insufficient to determine these to be disease genes and that the variants we identified cannot be referred to as pathogenic. In the results section we deliberately do not state whether these are plausible pathogenic alleles, merely we report them for their supportive role in the study and to stimulate their further investigation in the context of MAC. However, in the discussion we have downgraded the text where we have alluded to definitive pathogenicity, and removed references to clinical diagnostic panels, replacing these with research panels where relevant. In specific response to the reviewer’s comments on EPHB3, we have removed the following sentence from the discussion:

"Inclusion of EPHB3 on targeted gene panels for structural eye disease may help to improve diagnostic rates for MAC patients."

Additionally, we have updated Table 3 to include REVEL and PhyloP100 scores. We have also created 4x species amino acid conservation at the position of each variant (Figure S5)

2.7)       Genetic data for NID1 is more convincing with the exception of the Arg735His variant.  This one should be removed and placed in supplemental information.  Based on the discussion in the manuscript, the authors included it for completeness and generally agree with this assessment.    There is a high gnomad allele frequency and 36 homozygotes in gnomad suggesting this is likely to not cause such a severe phenotype in the patient.  For NID1 patients, since this has been associated with Dandy-Walker malformations a comment on whether these individuals were phenotyped for this malformation would be helpful.  As noted above, REVEL and molecular modeling may be helpful for further strengthening in silico pathogenicity. 

We are grateful the reviewer noted we were circumspect about the NID1 variant. However, we believe it is important to openly report all of these variants in the main manuscript for two reasons: (i) to ensure these genes are interrogated in patient DNA samples/panels within individual research laboratory contexts (i.e. MAC panels collected by clinicians whose sequence data is not publicly available), and (ii) to promote further functional work for these genes in the developing eye in model systems to determine their role in this process. Endeavours in these two areas will provide further evidence to determine whether they should be included in clinical diagnostic panels using targeted sequencing. In those laboratories where WGS is routinely used, we hope these genes are re-analysed bioinformatically to detect plausible variants for additional functional analyses.

We also appreciate the comment highlighting Dandy-Walker phenotyping. This term would have would have come up and been included in Table 3 for the output HPO terms of the patients in GEL if diagnosed, therefore this is not the case. We have added a comment indicating this in the legend for Table 3.

2.8)       The discussion is lacking in discussing the EPHB3 variants and potential pathogenicity.

To address this we have added the following comment into the discussion:

"The variants identified in this study cannot at this point be considered as pathogenic without further functional analyses. Mutations in EPHB3 have not yet been associated with defects in eye development, but the zebrafish CRISPR/Cas9 mutant phenotype data presented here together with its developmental expression in the ventral chicken eye tissue suggests this gene has an important role in normal oculogenesis and OFC and should warrant further study in this context."

2.9)       Given this is an evolutionary conservation based study, it would be interesting for the authors to report conservation of variants they observed in patients among the 4 species studied.

On the reviewer's suggestion we have added amino acid alignments to the manuscript in Supplemental Figure 5.

2.10)   Line 236 “expresison” should be expression

We have corrected this (Line 287 in revision).